# Destination Brand Experience: A Study Case in Touristic Context of the Peneda-Gerês National Park

Hugo Martins [1,*], Paulo Carvalho [2] and Nuno Almeida [3]

1   Department of Business Sciences, University of Maia, 4475-690 Maia, Portugal
2   CEGOT, FLUC, University of Coimbra, 3004-531 Coimbra, Portugal; paulo.carvalho@fl.uc.pt
3   CiTUR, ESTM, Polytechnic of Leiria, 2520-614 Peniche, Portugal; nunoalmeida@ipleiria.pt
*   Correspondence: hugomartins@ismai.pt

**Abstract:** Based on the scientific literature, this paper emphasises the destination brand experience (DBE) (multidimensional construct and second-order factor) in order to analyse the implications it plays regarding visitors' satisfaction, their intentions to revisit and their intentions to recommend it. In terms of methodology, a confirmatory factor analysis was used to test the model and the research hypotheses. The sample was composed of 507 tourists who visited the Peneda-Gerês National Park in Northern Portugal. Results showed an acceptable fit. The items of each construct were very strong. Positive significant results were found for all the considered hypotheses, particularly regarding the association of sensory DBE and behavioural DBE (subdimensions of the DBE scale) with satisfaction. The sensory DBE and affective DBE subdimensions of the DBE scale were meaningfully associated with visitors' intentions to recommend. Satisfaction was a strong mediator for sensory DBE impact on their intention to revisit and to recommend, and a less strong effect was found for satisfaction as a mediator for behavioural DBE impact on intentions to revisit and to recommend. The theoretical contribution of this study aimed to deepen the analysis of the DBE construct in its multidimensional aspect and its relationship with other constructs. The results are discussed in relation to their theoretical and practical relevance.

**Keywords:** destination brand experience; satisfaction; intentions to revisit and recommend; Peneda-Gerês National Park (PGNP)

## 1. Introduction

Tourism has been considered as one of the catalyst sectors in economic, social and cultural terms, generating wealth and employment for thousands of people. Furthermore, in recent decades, there has been a growing increase of tourist destinations which aim to position themselves in the market to generate competitiveness. According to [1], "tourism destination competitiveness is important for a destination to obtain a favourable position in the world tourism market and sustain a competitive advantage" (p. 257). This competitive pressure has been rising [2]. A touristic destination is "a physical space with or without administrative and/or analytical boundaries in which a visitor can spend an overnight. It is the cluster (co-location) of products and services, and of activities and experiences along the tourism value chain and a basic unit of analysis of tourism. A destination incorporates various stakeholders and can network to form larger destinations. It is also intangible with its image and identity which may influence its market competitiveness" [3] (p. 10). According to [4], "a brand becomes one of the most effective management instruments, including the development of marketing activities for both commercial enterprises and non-profit institutions as well as for entities undertaking territorial marketing activities" (pp. 215–216).

Marketers have realised that it is vital to study the way consumers experience brands so as to create and provide new brand experiences that are more appealing to consumers. As [5] mentions, experiences are interactions that occur between the brand (through

a service and/or a product) and the consumer, being a mix of physical performance and evoked emotions which are intuitively measured with the customer's expectations. Brands play an important role in connecting service providers with consumers and other stakeholders [6]. According to [7], to analyse consumer behaviour concerning destination brands, many marketing-related concepts have been applied, namely, destination image [8], destination (brand) personality [9] and destination (brand) identity [10]. The integrative models of brand and destination image present a practical consensus among researchers about the elements that form the global image of the destination, such as the existence of affective and personal values. However, there is no unanimity in the interrelation of the brand and the image of the destination.

Nevertheless, the widespread criticism that has been made of these studies on the application of these constructs is that they are incomplete or partial, as they do not cover all the experiences that impact consumers when they are stimulated by the brand [11]. The brand experience concept emerged as a response to these limitations in these studies, aiming to measure consumers' responses to a brand [7]. The brand has its own identity and it is a strong source of cognitive, sensory and affective relations, which result in brand experiences that are intended to be memorable and rewarding, i.e., the brand is an experience [11]. There is, therefore a need to understand the experience in the different stages of the decision-making process of tourist consumption. According to several authors, satisfactory experiences, whether associated with other factors or not, can function as appropriate measures to assess loyalty and consequent loyalty to the tourist destination [12–14]. Reference [15] even considered that, "given that the personal relevance of the destination in the choice process and the experience on site are likely to influence whether one recommends and returns to the same destination ( . . . ) likewise, high involvement in the destination experience can contribute to positive evaluations of destination attributes" (p. 12).

To manage a destination brand effectively, "destination marketers need to fulfil the wants/needs of destination customers" (p. 223) [6]. It is therefore necessary to implement and test brand experience construct scale in tourism settings. In this sense, the contribution of our article is to (re)apply a theoretical construct in destination brands, through the scale used by [7], in the touristic context; a scale previously created and validated by [11]. The territory chosen to test this scale was a relevant tourist destination in north Portugal and the most prestigious protected area in the country: the Peneda-Gerês National Park (PGNP). PGNP has a long heritage of a historical and cultural nature (historic villages, megaliths, Celtic and Roman remains, medieval and modern castles and stone pillories, and the *Espigueiros* (grain houses) of Soajo and Lindoso, among others). In recent decades, there has been a high growth of tourist demand in this particular region, resulting from the projection of the Gerês brand internationally, according to the Nature and Forest Conservation Institute [16].

Therefore, we intend to investigate the importance of the DBE, a multidimensional construct, regarding visitor satisfaction, as well as intentions to recommend and to return to the destination. This study assumes greater importance at a time when there is the possibility of facing an evolution in the tourism paradigm. This evolutionary perspective has been increasingly used in the social sciences. By analysing this perspective, we intend to explore new areas of knowledge around subjects that have already been tested in the tourism context, such as DBE. Furthermore, so that the DBE construct will not be compromised by exogenous variables such as pandemics, it would be desirable to pay particular attention to the reaction of tourists to this construct. Consequently, it is vital to understand how a destination brand is experienced by tourists [6].

## 2. Literature and Hypotheses

### 2.1. Brand Experience: Concept and Characteristics

Consumers seek brands that offer memorable and unique experiences [17]. This involvement with the brand arises due to the consumers' needs, values and interests [11].

When acquiring a service and/or a product, the consumer is exposed not only to its functional attributes, but also to a series of "brand-related stimuli such as colours, shapes, fonts, designs, slogans" [11] (p. 54), among others. These stimuli are part of the brand identity and the brand communication strategies in environments where the brand is sold or advertised.

Despite being related, the brand experience construct is conceptually distinct from the other brand-related constructs, like brand engagement, brand attitudes, brand attachment, brand personality and customer satisfaction. It is thus a more comprehensive concept, which offers us a more holistic view [18]. The development of experienced branding must consider the basic process of the branding. In other words, it must be a virtuous circle. The effects of the brand are measured in the "final" stage and under this evaluation the basis must receive improvements. In this way this process will start again in the "new first stage" in which the brand is re-examined with newly relevant elements, according to the World Tourism Organization and European Tourism Commission [19] (p. 37). There are already different points of view regarding the definition of brand experiences in the literature, and study of "brand experience is a promising field" [6] (p. 232). As stated by [20], brand experience is the way customers use the brand, express themselves about it, and search for brand information, events, promotions, etc. The most consensual definition of brand experience among researchers, and the most referenced and cited in the literature, is "sensations, feelings, cognitions and behavioural responses evoked by brand-related stimuli that are part of a brand's design and identity, packaging, communications and environments" [11] (p. 52). These authors conceptualized and delimited the brand experience phenomenon, considering that the consumers' responses can be behavioural as well as subjective and internal (feelings, sensations and cognitions). Both types of responses are aroused by stimuli that are part of the brand, its identity (name, logo, signature) and design, communications (advertising, brochures, website), packaging and environment (shops, events) [11].

To study this phenomenon, [11] sought to identify the subdimensions of this new construct and create a measurement scale. A scale within the scope of brand experience would necessarily have to cover a vast literature, so that its items would be commonly accepted, in the most diverse areas such as philosophy [21], cognitive sciences [22], management and experience marketing [23,24]. The authors created a scale that could assess the consumers' brand experience in sensory, affective, intellectual, behavioural or social terms, and not merely measure a specific experience. According to Brakus et al. [11], although there are already some important scales that can measure these parameters, the specific components on these scales are incomplete. In fact, the brand experience scale aims to measure responses in a more holistic way, assessing the consumers' sensory, affective, behavioural, intellectual or social experiences. Effectively, the stimuli provided by the brand experience to the consumer, in the form of sensations and feelings, provokes responses that are both holistic and subjective [25]. In short, [11] sought to try a new scale for this construct, which was developed from a set of six studies carried out for this purpose. This scale was used for the first time in 2014, applied in the context of tourism, and it served as a matrix for the present investigation.

The scale created is based on a set of four dimensions: (a) a sensory dimension that refers to the efforts developed by marketing to appeal to the human senses, through hearing, sight, touch, taste and smell [26]; (b) an affective dimension that manifests itself through the consumers' feelings and/or thoughts, aiming the creation of affective experiences that vary their strength depending on how the consumer relates to the brand [27]; (c) a behavioural dimension which seeks, through body experiences, lifestyles and interactions, to enrich consumers' lives, showing them other ways of getting alternative lifestyles and different interactions; and (d) an intellectual dimension which appeals to the consumers' creativity and innovation, as it has the ability to promote cognitive experiences so that they get involved with a brand creatively, which helps them develop feelings such as surprise and admiration. These dimensions can be evoked by brands individually or in groups. In



this sense, a company that wants to provide a good experience to its consumers must first formulate a mental model concerning the areas that can affect the consumer's senses. In fact, there are companies whose primary objective is to provide experiences, for example Odisseias, Virgin Experience Days and Buyagift, which are the major brands in the market selling experience vouchers. Another example of companies that sell experiences is the case of Starbucks, which sells not only coffee, but also an experience around the consumption of coffee itself [28].

As stated by [11] "brand experiences vary in strength and intensity; that is, some brand experiences are stronger or more intense than others. As with product experiences, brand experiences also vary in valence; that is, some are more positive than others, and some experiences may even be negative. Moreover, some brand experiences occur spontaneously without much reflection and are short-lived; others occur more deliberately and last longer" [11] (p. 53). As time goes by, enduring brand experiences stored in the consumers' memory may influence their satisfaction and their loyalty [29]. In the opinion of [11], experiences can also occur in an indirect way, for example, through advertising, marketing communications or websites. They can also happen in an unexpected way, as they can occur when consumers express no interest in them or do not form a strong bond with the brand. The brand experience is a personal information provider that can be used in future decision-making processes, such as a new purchase intention [30]. Further, [31] considers that brand loyalty develops through repeated purchase experiences of a brand over time.

As today's markets are highly competitive, consumers themselves have become less tolerant of any problems related to what has been promised by the brand and what it actually provides [32]. In the face of a possible shortcoming or failure of a brand, the consumer will quickly look for another competing brand. According to [32], if the brand experience meets the consumer's expectations, there may be co-creation of value. Consequently, many companies design their strategies around brand experiences, providing their target consumers with sensory, emotional, cognitive, behavioural and relational values. All these factors complement and enrich the brand's functional values, thus enhancing its value.

Corroborating [33], the experience becomes a brand image, "forming the mental conceptions and perceptions of interactions and inputs in the service process, which constitutes the final outcome of the multi-sensory experience within a brand perspective" (p. 263). This point of view is defined as an individual's feelings, beliefs, opinions and thoughts about a certain brand, based on general experience [11,34]. Given the characterisation of this recent construct and its importance in companies that want to gain a place in the global market, it was important to test this construct with its dimensions in the tourism context. This study also intends to prove the consistency of the DBE scale and provide the Destination Marketing Organization, who seek to add value to their brand, a significant orientation in order to discover ways to improve the brand experience and, consequently, visitor loyalty. It is therefore crucial to realize the potential it can offer marketers "in both increasing the perceived value of their current product offerings as well as their brand equity" [35] (p. 141). In addition, this study reveals the importance that destinations promote themselves, considering thei branding strategies, to consolidate and create the idea that through positive experiences, tourist destinations can differentiate themselves.

The use of a brand has been a constant practice on the part of organizations, and places/territories also started to adopt this behaviour. Tourist destination managers consider brands from a perspective of value creation, through the adoption of differentiation strategies of branding, in an increasingly competitive market [36]. Therefore, companies and destinations are on the same level, as both intend to take advantage, through their identity, of their competitive advantages in order to improve their positions in the market, since "places can be easily assumed having the characteristics identity, differentiation and personality and therefore can be managed to maximize equity, value and awareness" [37] (p. 510). The authors of [38] consider that tourist destinations have similar properties to products and services, as both have tangible and intangible attributes. However, the prac-

tice of destination branding is much more complex because a tourist destination is limited by attributes that are difficult to control, while the branding of a product is dependent on the company's survival [39]. In this sense, for the destination brand to be successful, it is important that there is involvement by the destination's stakeholders [36]. Destinations came to be understood as true brands, being managed strategically [40]. The mark of destiny is a consistent mixture of elements that distinguish this destination from other competitors through experiences that are intended to be memorable.

### 2.2. Brand Experience Applied to Tourism

There is a vast academic literature on branding and brand consumption, as branding can be a means of differentiating services and products [41–44]. The brand associated with a tourist destination can also be a means of differentiation and an advantage in terms of competitiveness, according to several authors [45–47]. The literature itself considers it is important to be aware of the way a destination brand is experienced by tourists [6,48,49].

In a tourism context, the DBE construct may solve any problems that may arise in the existing scales, thanks to its comprehensive features. This construct offers a more holistic perception of the destination brand, offering a more complete evaluation of the brand itself in affective, sensory, intellectual and behavioural terms. As claimed by [7], while brand attitudes are general evaluations, brand experiences "include specific sensations, feelings, cognitions, and behavioral responses triggered by specific brand stimuli" (p. 124).

The first time this brand experience construct was applied to the tourism destination was in the study developed by [7], where they tested the marketing scale used by [11], and considered that this scale provided "a unified, formal, rigorous and systematic model that captures the four key dimensions of destination brand experience" [7] (p. 124).

In order to test the scale readapted to the tourism context in a global way, the authors sought to determine which construct dimensions had the greatest impact on visitors, using a research model and a variety of destinations. The authors of [7] concluded that visitors are mainly influenced by sensory experiences, which suggests the satisfaction of hedonistic needs. Some studies suggest a holistic approach, exploring the impact of multi-sensory experiences on satisfaction and loyalty [7,50,51]. IOn the other hand, the role of extraordinary sensory experiences in shaping a destination brand can suggest that destination marketing organizations (DMO) should start from the five senses to develop the tourist-brand relationship [51]. Therefore, "the effective way to make tourists fall in love with a destination brand is by providing extraordinary sensory experiences rather than common sensory experiences", according to [51] (p. 188). However, despite the prominence of sensory experiences, their study reveals that affective experiences are very important in certain circumstances. As a result, "travel agents and tourism providers should focus more on sensory aspects of visits and ( . . . ) design tourism experiences from a sensory and affective perspective" [7] (p. 137). They also add that behavioural and intellectual experiences seem to be harder to accomplish in a tourism context. Similarly to [52], the authors concluded that the intervening entities linked to the tourism sector should give special focus to sensory experiences, highlighting touch, images, sounds, tastes and smells. This suggests that "the design of tourist experiences should provide scope for individuals to learn and to be challenged, and to develop new, social perspectives on life" [7] (p. 137). They suggest that the profile of the DBE is likely to differ in conformity with the place characteristics, and that the experiences should be carefully selected. They also assume that tourism specialists should focus on a comprehensive profile of diverse tourists in order to comprehend how a destination brand is experienced in its multiple subdimensions. Additionally, they consider that the DBE is a meaningful determinant in attracting tourists to that destination and that the degree of satisfaction plays a vital role in tourist experiences.

On the assumption that brand experience can apply to all categories of services and products, even in tourism [53], and that experience provides value, we share the same thought as [11] that the more "a brand evokes multiple experience dimensions,

and therefore has a higher overall score on the scale, the more satisfied a consumer will be with the brand" (p. 63). Furthermore, [11], through their studies, concluded that there is a connection between an enjoyable brand experience, customers' loyalty to the brand and satisfaction, and that brand experiences, by creating positive outcomes, will affect consumers' decision processes, in the sense that they are more likely to repeat purchases and to make recommendations to other people. However, to be able to explain consumers' decision-making process, as well as their satisfaction and loyalty, it is important to study all the aspects that guide this process. The scale created by [11] has filled a gap identified in previous models, since it is a more holistic model that integrates the intellectual approach (cognitive model) and the affective approach (emotional model), while including a behavioural and sensory component.

Regarding destination branding, there is already some scientific literature, namely [9,10,13,46,54], positively relating destination branding to brand satisfaction and loyalty. Loyalty, according to [13,55], is usually measured through two constructs: visitors' intention to revisit a destination, and visitors' desire to recommend that destination to someone else. We understand that this research is an opportunity to test this scale in order to understand the four components that [11] indicates as brand experience subdimensions. Therefore, we aim to test DBE in a certain tourist destination with specific characteristics, relating this construct to the visitors' satisfaction and their intentions to recommend and revisit that place.

In order to contribute to this theoretical model, we have formulated the following hypotheses:

**Hypotheses 1 (H1).** *A positive (H1a. sensory, H1b. affective, H1c. behavioural and H1d. intellectual) DBE will increase visitor satisfaction with the destination.*

**Hypotheses 2 (H2).** *A positive (H2a. sensory, H2b. affective, H2c. behavioural and H2d. intellectual) DBE will increase visitor intention to recommend the destination.*

**Hypotheses 3 (H3).** *A positive (H3a. sensory, H3b. affective, H3c. behavioural and H3d. intellectual) DBE will increase visitor intention to revisit the destination.*

The concept of satisfaction may be understood as the evaluation of the relationship between: (1) the expectations the consumer has about a given product or service, and (2) the performance perceived after buying that service or product. Thus, satisfaction corresponds to the degree to which the consumers feel fulfilled when inferring which characteristics of a product or service pleased them more during the consumption experience [28]. According to marketing studies on various types of brands, it can be noticed that the most satisfied consumers tend to buy the brand's products or services again and tend to recommend their purchases to friends and acquaintances, who will then become new customers of that brand [56].

Research articles by [57–59] indicate that there is a direct and positive effect of satisfaction on repurchase intentions and recommendations by word-of-mouth. In fact, there are already several studies on theory and empirical evidence relating satisfaction with behavioural intentions to revisit and recommend, such as those of [13,55,60], among others. In fact, these studies show that tourists are highly predisposed to revisit the destination after a positive experience that met and/or exceeded their initial expectations. Reference [13] corroborates this idea by considering that "tourists' positive experiences of service, products, and other resources provided by tourism destinations could produce repeat visits as well as positive word-of-mouth effects to friends and/or relatives" (p. 625). Furthermore, those who are more likely to return to the destination also suggest it to family and friends, because "recommendations by previous visits can be taken as the most reliable information sources for potential tourists. Recommendations to other people (Word of Mouth, WOM) are also one of the most often sought types of information for people interested in travelling" [13] (p. 625).

Consequently, we have formulated the following hypotheses:

**Hypotheses 4 (H4).** *Higher visitor satisfaction will positively influence visitor intention to recommend the destination.*

**Hypotheses 5 (H5).** *Higher visitor satisfaction will positively influence visitor intention to revisit the destination.*

**Hypotheses 6 (H6).** *The relationship between (H6a. sensory, H6b. affective, H6c. behavioural and H6d. intellectual) DBE and visitor intention to recommend the destination will be positively mediated by visitor satisfaction with the destination.*

**Hypotheses 7 (H7).** *The relationship between (H7a. sensory, H7b. affective, H7c. behavioural and H7d. intellectual) DBE and visitor intention to revisit the destination will be positively mediated by visitor satisfaction with the destination.*

It is expected that, "after a positive experience at a particular destination where, in general terms, expectations have been met and in some cases even exceeded, tourists experience a feeling of overall satisfaction" [61] (p. 75). As stated by the same author, the reliability of the destination also contributes for the creation of this feeling of overall satisfaction, namely if that destination met the consumers' initial expectations and the predictability of the destination (minimization of risks and uncertainties), leading to significant levels of confidence in that destination. If a given tourist destination is highly trusted, tourists will be more likely to develop behavioural intentions to revisit and recommend it.

*2.3. Peneda-Gêres National Park Characteristics*

Protected areas correspond to a significant variety of designations, typologies, geographical environments and management models [62,63] and are increasingly emerging as highly relevant tourist destinations [64,65]. Simultaneously, tourists are looking for personalized activities and experiences of great symbolic value, mostly outdoors, such as hiking, cycling, fauna/flora watching and water activities [66–68].

Located in the northwest of Portugal, the PGNP occupies an area of approximately 703 square kilometres and it is divided into eighteen parishes belonging to five municipalities (Terras de Bouro, Montalegre, Melgaço, Arcos de Valdevez and Ponte da Barca), inhabited by 6383 people in 2019 (according to the National Institute of Portuguese Satistics). It a mountainous area with different types of relief that can reach an altitude of more than 1500 m. It also has natural habitats that support a rich and varied fauna and flora with several endemic, rare or endangered species. Therefore, this region is highly noteworthy at the national and international levels. Despite being closely associated with nature tourism products, the PGNP also offers a huge variety of material and immaterial cultural heritage, which makes tourism a pillar of enormous relevance for the development of the region (Figure 1).

Since 1997, this protected area, together with the Spanish Natural Park of Baixa Limia-Serra do Xurés, form the Gerês-Xurés Transfrontier Park, which, in 2009, was considered a World Biosphere Reserve (Transfrontier Biosphere Reserve "Gerês-Xurés") by UNESCO. In the context of the European Union, it is part of the Natura 2000 network and it was recognised as "a Site of Community Importance (defined in the European Commission Habitats Directive)" [36] (p. 193). In 2010, the International Year of Biodiversity, the PGNP was considered as one of the seven Natural Wonders of Portugal, under the category of Protected Areas.

At the international level, the PGNP sought to be part of several highly prestigious and respected networks. For instance, it belongs to the Network of Biogenetic Reserves of the Council of Europe because of the area "Matas de Palheiros-Albergaria." It is also part of the Federation of Nature and National Parks of Europe, as well as the PAN Parks Foundation. Being a partner in the PAN Parks Foundation allows the PGNP to be part of a network of excellence where only the best parks in Europe are listed. Moreover, the PGNP is the only park in the Iberian Peninsula belonging to this network. With the certification granted by the PAN Parks Foundation, a massive influx of foreign tourists was expected, namely from northern Europe, since the PGNP was included in the itinerary of large tour operators

specializing in nature tourism. Indeed, there has been a rise in the number of tourists to these areas that offer recreational and leisure activities in direct contact with nature and local cultures. That is why these areas have become new tourist destinations [70].

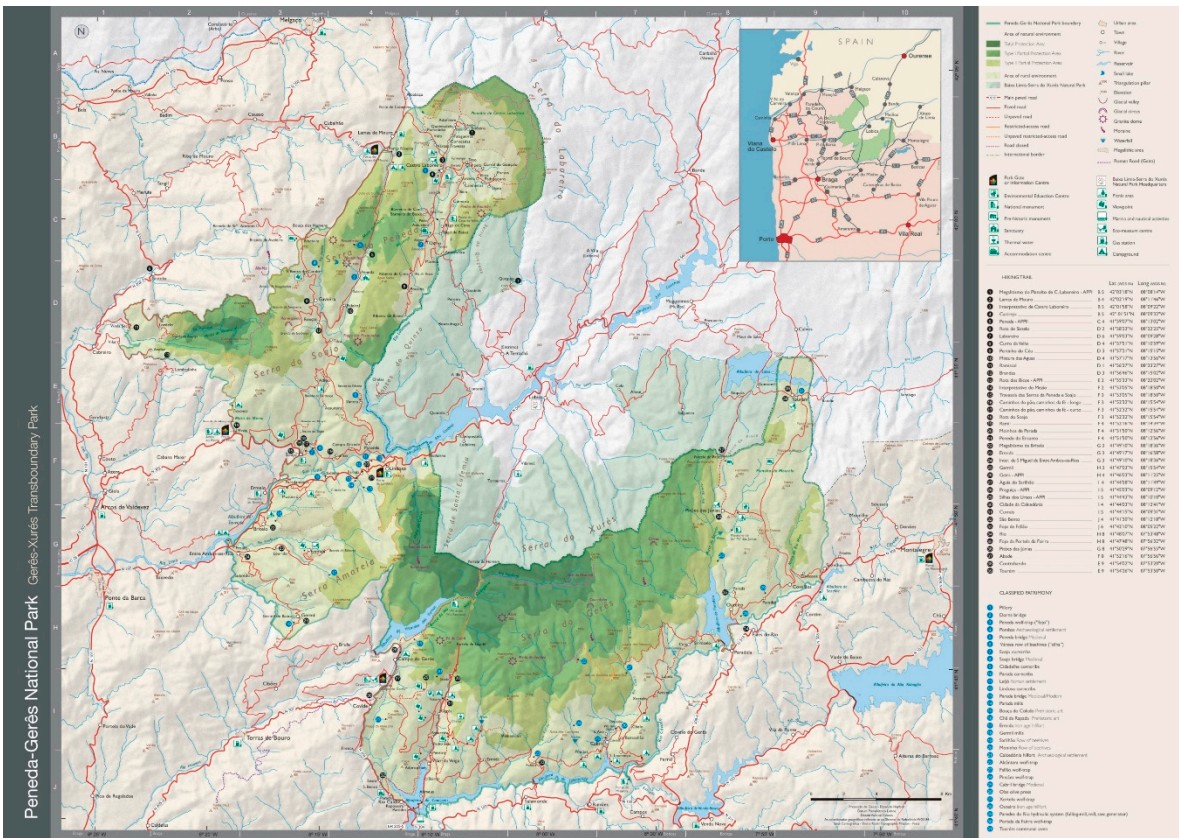

**Figure 1.** Peneda-Gerês National Park map. Source: [69].

Through the data provided by the Institute for Nature Conservation and Forests, it is possible to observe the evolution of the number of visitors to the main protected areas (Figure 2). In the case of the PGNP, from 2010 to 2019, there was a growth of 114%, and that the number of 100,000 visitors was surpassed in 2016. However, from 2017 to 2019, there was a decrease in the number of visitors (from 115,804 in 2017 to 103,593 in 2019). It should be considered that until 2009, only the contacts with the entities managed by the ICNF were considered. Since the beginning of the 21st century, the five PGNP gates were open officially, working as a boosting factor for tourism, so since 2010 entries through the gates have also been considered.

Additionally, it is also possible to observe that other protected areas have been enjoying notability, namely the Sado Estuary Natural Reserve, which reached 82,242 visitors in 2019. Another protected area that is having an increase in the number of visitors is the Ria Formosa Natural Park, which attracted up to 60,061 visitors in 2019. We can also note that the data regarding the Serra da Estrela Natural Park has fluctuated greatly, reaching a peak of visitors in 2019 (18,429) (Figure 2). In the case of the Serras de Aire e Candeeiros Natural Park, the number of visitors decreased between 2009 and 2014. Since then, the number of visitors rose, as there was a greater emphasis on the creation of various types of visits to the Pegadas de Dinossáurios da Serra de Aire Natural Monument.

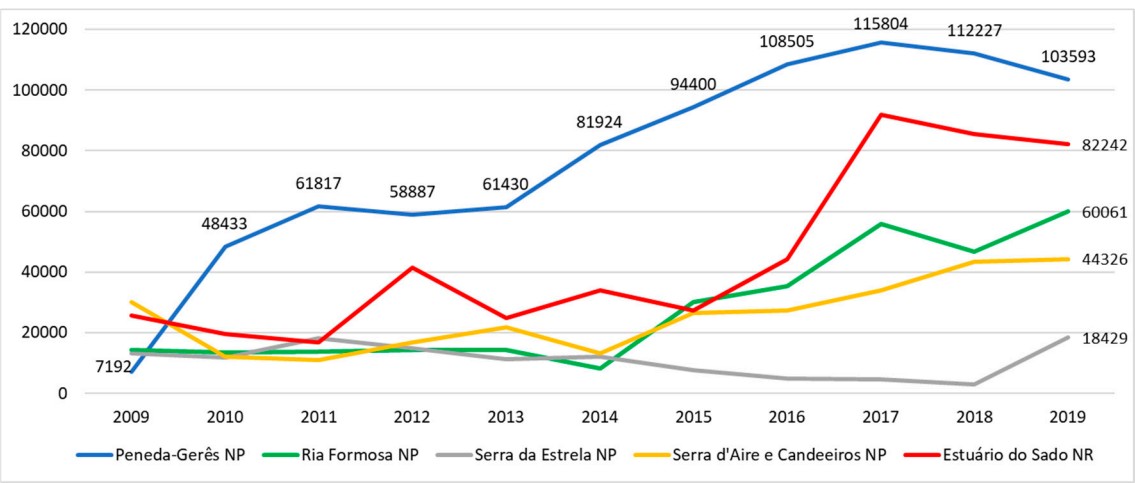

**Figure 2.** Visitors in the main Portuguese protected areas (2009–2019). Source: own illustration, based on Reference [70] (adapted).

## 3. Methodology

Focussing on the literature review, we elaborated a research model that presents the relationships between the constructs. From the proposed model, we formulated the seven hypotheses referred to previously. The research model and the hypothetical connections were tested using data collected from tourists at PGNP, the only Portuguese national park, located in northern Portugal. PGNP also stands out as having a network of infrastructure that provide conditions to attract tourists. Besides, it offers many tourist products such as nature, health and well-being and it promotes religious, nautical and cultural tourism.

Therefore, we intend to test the relationships between DBE as a multidimensional construct, tourists' satisfaction and their intentions to recommend and revisit the destination. Our empirical study relied heavily on fieldwork. This methodology seemed adequate since we aimed to obtain information to confirm results identified in the literature review we carried out, concerning the constructs we intended to analyse.

The fieldwork took place between June and October 2016. The target population was tourists who stayed overnight in the PGNP. The sample was intended to be representative so that it would be possible to draw and extrapolate conclusions [71].

The technique chosen was the questionnaire survey, made available in four languages (Portuguese, English, French and Spanish) in order to get the opinion of national and foreign tourists who visited this tourist destination.

The questionnaire was structured with closed-ended questions. It focussed on the brand experience scale used by [11], which was previously adapted by [7] to the subject under investigation. This scale comprises four brand experience subdimensions: affective, sensory, intellectual and behavioural. Additionally, the questionnaire also allowed us to collect satisfaction measure items ("I believe I did the right thing when I chose to visit the PGNP; I am happy about my decision to visit the PGNP; Globally, I'm satisfied with the PGNP as a tourist destination"), a single item to revisit the destination ("I will revisit the PGNP again") and a single item to recommend it ("I will recommend the PGNP to my friends and relatives"). Construct measurement was performed through an interval attitude scale in the Likert interval format, expanded to seven points. Qualitative variables (nominal and ordinal) were also used to assess the tourists' socio-demographic profile and information about their stay. The scales were designed to be as robust and consistent as possible, based on adaptations of tested scales, in some cases by more than one author, which allowed us to check the previously formulated hypotheses.

Since the target population was the tourists who stayed overnight in the accommodation units within the PGPN, the sample was non-probabilistic by convenience. In order to carry out the empirical study, receptionists from local accommodation units and tourism

enterprises were asked to collaborate in the delivery of the questionnaire. After the data collection, the questionnaires were coded and validated. The sample was significant [71], with a total of 507 respondents.

Statistical analysis was performed on XLSTAT. Continuous variables were described as means (M) and standard deviations (SD). Absolute (n) and relative (%) frequencies were calculated for categorical variables. Standard errors (SE) and critical ratios (CR) were calculated to assess study hypotheses. A confirmatory factor analysis (CFA) was implemented concerning reliability and validity (Tables 2 and 3). We measured composite reliability (CR), based on CR > 0.7 [72] and converging validity with average variance extracted (AVE), considering > 0.50 [73]. To assess discriminant validity, we calculated squared intercorrelations and compared them with the AVE of the constructs [73].

Direct and indirect (mediated) estimated effects are presented in Tables 3 and 4. Model quality was assessed with the R2 and F tests. The significance of the associations between the constructs was assessed with Pr > |t|; f2, and used to measure effect size of each direct effect.

Goodness of fit indices were calculated to assess the research model fit, considering as main criterion the relative goodness of fit above 0.90 [74,75]. The significance threshold was $p < 0.05$.

## 4. Results and Discussion

The sample was composed of 507 respondents: 259 (51.1%) male and 248 (48.9%) female. The participants were aged between 18 and 80 years (M = 38.45; SD = 13.38); 298 (58.8%) were married, 179 (35.3%) single and 30 (5.9%) divorced/widowed. Most participants (n = 438; 86.4%) were from Portugal.

Tables 1 and 2 show results of validity and reliability of the DBE construct established on a CFA. The results of CR were all considerably above 0.70, sensory DBE (0.852), affective DBE (0.801), behavioural DBE (0.781) and intellectual DBE (0.793). The items of each construct were very strong ($p < 0.001$).

**Table 1.** Psychometric analysis of the DBE construct.

| | Item | M | SD | Loading | SE | CR |
|---|---|---|---|---|---|---|
| Sensory DBE (AVE = 0.772; CR = 0.852) | 1. PGNP makes a strong impression on my senses, visually and in other ways. | 6.034 | 0.980 | 0.847 | 0.027 | 31.746 *** |
| | 2. I find PGNP interesting in a sensory way. | 6.095 | 0.899 | 0.921 | 0.010 | 89.390 *** |
| | 3. PGNP appeals to my senses. | 5.955 | 0.989 | 0.867 | 0.016 | 55.578 *** |
| Affective DBE (AVE = 0.715; CR = 0.801) | 4. PGNP induces feelings and tranquillity. | 6.178 | 0.915 | 0.846 | 0.017 | 48.570 *** |
| | 5. I do have strong emotions for PGNP. | 5.548 | 1.127 | 0.843 | 0.018 | 45.871 *** |
| | 6. PGNP is an emotional area. | 6.018 | 0.992 | 0.848 | 0.016 | 53.506 *** |
| Behavioural DBE (AVE = 0.696; CR = 0.781) | 7. I engage in physical activities and behaviours when I am on PGNP. | 5.499 | 1.238 | 0.823 | 0.022 | 37.203 *** |
| | 8. PGNP gives me bodily experiences. | 5.809 | 1.044 | 0.868 | 0.014 | 62.113 *** |
| | 9. PGNP is activity-oriented. | 5.661 | 1.154 | 0.810 | 0.023 | 35.521 *** |
| Intellectual DBE (AVE = 0.721; CR = 0.793) | 10. I engage in a lot of thinking when I am on PGNP. | 4.523 | 1.731 | 0.932 | 0.008 | 121.496 *** |
| | 11. PGNP makes me meditate. | 4.554 | 1.710 | 0.945 | 0.006 | 160.793 *** |
| | 12. PGNP stimulates my curiosity and problem-solving capacities. | 5.704 | 1.195 | 0.633 | 0.031 | 20.417 *** |

*** $p < 0.001$.

**Table 2.** Test for discriminant validity (squared correlations < AVE).

|  | 1 | 2 | 3 | 4 | 5 | 6 | 7 |
|---|---|---|---|---|---|---|---|
| 1 Sensory DBE | 1 | | | | | | |
| 2 Affective DBE | 0.649 | 1 | | | | | |
| 3 Behavioural DBE | 0.311 | 0.386 | 1 | | | | |
| 4 Intellectual DBE | 0.161 | 0.216 | 0.180 | 1 | | | |
| 5 Satisfaction | 0.425 | 0.348 | 0.260 | 0.091 | 1 | | |
| 6 Recommend (WOM) | 0.443 | 0.391 | 0.215 | 0.124 | 0.580 | 1 | |
| 7 Revisit (REV) | 0.287 | 0.296 | 0.172 | 0.079 | 0.386 | 0.634 | 1 |
| AVE | 0.772 | 0.715 | 0.696 | 0.721 | 0.817 | | |

Through the results obtained from the DBE scale items, it is possible to observe that there four items stood out with an average higher than all others, standing above the value six (agree) of the Likert scale: items 1 ("PGNP makes a strong impression on my senses, visually and in other ways") and 2 ("I find PGNP interesting in a sensory way"), from the sensory subdimension; and items 4 ("PGNP induces feelings and tranquility") and 6 ("PGNP is an emotional area") from the affective subdimension.

However, it is possible to observe that two items had a lower average than all the others, ranging between point four (neither satisfied nor dissatisfied) and point five (satisfied) on the Likert scale: items 10 ("I engage in a lot of thinking when I am on PGNP") and 11 ("PGNP makes me meditate"), both from the intellectual subdimension. The remaining items are between points five (satisfied) and six (very satisfied) (Table 1).

On the one hand, this information corroborates studies that claim that not all aspects would be significant, and may vary according to the territory or even the specific characteristics of the experience (e.g., [7,35]). However, there are studies in which the affective and sensory subdimensions are not as valued and do not have a great impact, with an appreciation of the other subdimensions (e.g., [76]). Therefore, it is possible to understand that the experiences that cause the greatest impact on tourist destinations such as protected areas are more sensory and affective experiences, corroborating studies such as Barnes et al. (2014). It is therefore important that the agents responsible for tourism at the PGNP invest and focus more on this type of experience. It is important to make a strong impression in sensory (e.g., [77]) and affective terms.

Convergent validity measured with AVE ranged from 0.696 to 0.772, robustly above the recommend of 0.50. All items were strong in their DBE sub-dimension, and no evidence of crossloadings was found. The AVEs for the different constructs were considerably larger than the squared intercorrelations.

The results of the research model testing using PLS path modelling, are shown in Table 3, which presents hypotheses 1 to 5. Overall research model fit was considered to be good with relative GoF of 0.957, clearly above the 0.90 threshold. Table 3 shows significant results for all the considered hypotheses, particularly regarding the association of the sensory DBE and the behavioural DBE items of the DBE scale with satisfaction (H1a: β = 0.473, $p < 0.001$; H1c: β = 0.193, $p < 0.001$).

The sensory DBE and affective DBE items of the DBE scale were considerably associated with the intention to recommend (H2a: β = 0.182, $p < 0.001$; H2b: β = 0.146, $p < 0.001$).

Of all the DBE scale subdimensions, the relationship between the sensory subdimension and satisfaction stands out the most. It is therefore important that the entities responsible for the PGNP focus on experiences linked to the senses.

This reveals that sensory experiences exert a strong influence, being determinant not only in satisfaction but also in revisiting and recommending, in line with studies by [7,77].

**Table 3.** Testing of the research model.

| Relationship | Estimate | SE | t | Pr > \|t\| | $f^2$ |
|---|---|---|---|---|---|
| Sensory DBE → Satisfaction | 0.473 | 0.056 | 8.476 | <0.001 | 0.143 |
| Affective DBE → Satisfaction | 0.096 | 0.060 | 1.587 | 0.113 | 0.005 |
| Behavioural DBE → Satisfaction | 0.193 | 0.043 | 4.482 | <0.001 | 0.040 |
| Intellectual DBE → Satisfaction | −0.014 | 0.038 | −0.371 | 0.710 | 0.000 |
| Satisfaction: $R^2$= 0.459; F = 106.534; *p* < 0.001 | | | | | |
| Sensory DBE → Recommend | 0.182 | 0.049 | 3.735 | <0.001 | 0.028 |
| Affective DBE → Recommend | 0.146 | 0.049 | 2.957 | 0.003 | 0.017 |
| Behavioural DBE → Recommend | −0.038 | 0.036 | −1.060 | 0.290 | 0.002 |
| Intellectual DBE → Recommend | 0.059 | 0.031 | 1.920 | 0.055 | 0.007 |
| Satisfaction → Recommend | 0.558 | 0.036 | 15.321 | <0.001 | 0.469 |
| WOM: $R^2$ = 0.640; F = 178.459; *p* < 0.001 | | | | | |
| Sensory DBE → Revisit | 0.052 | 0.061 | 0.857 | 0.392 | 0.001 |
| Affective DBE → Revisit | 0.225 | 0.062 | 3.643 | <0.001 | 0.026 |
| Behavioural DBE → Revisit | 0.013 | 0.045 | 0.300 | 0.764 | 0.000 |
| Intellectual DBE → Revisit | 0.015 | 0.039 | 0.394 | 0.694 | 0.000 |
| Satisfaction → Revisit | 0.443 | 0.046 | 9.699 | <0.001 | 0.188 |
| Intention to Revisit: $R^2$ = 0.435; F = 77.218; *p* < 0.001 | | | | | |
| Goodness of Fit Index | GoF | GoF Bootstrap | | SE | CR |
| Absolute | 0.617 | 0.621 | | 0.027 | 22.572 *** |
| Relative | 0.957 | 0.944 | | 0.025 | 37.977 *** |
| Outer model | 0.999 | 0.997 | | 0.021 | 47.010 *** |
| Inner model | 0.958 | 0.947 | | 0.015 | 62.135 *** |

*** *p* < 0.001.

A relationship that was not verified was between intellectual DBE and satisfaction, showing that not all sub-dimensions are significant, especially in tourist destinations such as protected areas. In reality, this type of relationship stands out more in products/services than in tourist destinations (e.g., [78]). In addition, "sensory impressions ( ... ) provides incremental explanatory power on loyalty" [79] (p. 1). With this we recommend that destination marketers and travel intermediaries such as travel agents and tour operators, should promote the emotional experience of the destination in their advertising campaigns. Satisfaction was also found to have a significant impact on the intention to recommend (H4: β = 0.558, *p* < 0.001), already corroborating several studies (e.g., [61,80]).

Affective DBE, a component of the DBE scale (H3c: β = 0.225; *p* < 0.001), and satisfaction (H5: β = 0.443; *p* < 0.001) were found to have a significant impact on the intention to revisit.

Furthermore, considerant proportions of variance in the outcome measures are clarified by the research model, with 45.9% of satisfaction ($R^2$ = 0.459; F = 106.534; *p* < 0.001), 64.0% of intention to recommend ($R^2$ = 0.640; F = 178.459; *p* < 0.001) and 43.5% of intention to revisit ($R^2$ = 0.435; F = 77.218; *p* < 0.001) (Table 3).

Table 4 presents mediated effects through satisfaction in the research model, testing hypotheses 6 and 7. The mediating effect of satisfaction on the relationship between DBE and both the intention to revisit and the intention to recommend was found to have support on sensory and behavioural DBE. In particular, satisfaction was a strong mediator for sensory DBE impact on the intention to recommend (H6a: β = 0.264, *p* < 0.001) and the intention to revisit (H7a: β = 0.210, *p* < 0.001). This suggests that satisfaction cannot be seen as an end in itself, but as an attribute in the process of creating a consistent DBE, capable of retaining visitors. Although much of the literature considers emotional and cognitive experiences to be fundamental in influencing visitor satisfaction and loyalty [12], our study highlights sensory experiences.

**Table 4.** Mediated effects through satisfaction in the research model.

| IV | DV | Effect | SE | Z | *p* |
|---|---|---|---|---|---|
| Sensory DBE | Intention to Recommend | 0.264 | 0.268 | 5.749 | <0.001 |
| Affective DBE | Intention to Recommend | 0.053 | 0.056 | 1.227 | 0.110 |
| Behavioural DBE | Intention to Recommend | 0.108 | 0.106 | 3.853 | <0.001 |
| Intellectual DBE | Intention to Recommend | −0.008 | −0.009 | −0.333 | 0.630 |
| Sensory DBE | Intention to Revisit | 0.210 | 0.207 | 4.800 | <0.001 |
| Affective DBE | Intention to Revisit | 0.042 | 0.044 | 1.168 | 0.121 |
| Behavioural DBE | Intention to Revisit | 0.085 | 0.082 | 3.488 | <0.001 |
| Intellectual DBE | Intention to Revisit | −0.006 | −0.007 | −0.348 | 0.636 |

A moderate effect was also found for satisfaction as a mediator for behavioural DBE impact on the intention to recommend (H6c: β = 0.108, $p < 0.001$) and the intention to revisit (H7c: β = 0.085, $p < 0.001$). In fact, there are some activities in the PGNP that allow some exercise, namely hiking or recreational activities. In the territory of the PGNP, there are 40 trails/paths, as well as the 12 routes with cartographic or GPS guidance that visitors can enjoy.

A weaker effect was found for satisfaction as a mediator for affective DBE impact on the intention to recommend (H6b: β = 0.053) and intellectual DBE impact on the intention to recommend (H6d: β = −0.008). A less strong effect was also found for satisfaction as a mediator for affective DBE impact on the intention to revisit (H7b: β = 0.042) and intellectual DBE impact on the intention to revisit (H7d: β = −0.006). The intellectual DBE does not reveal a great impact on recommendation and revisit, most likely because of the type of tourist destination where nature tourism is what attracts visitors. If the territory under analysis were a city with literary or even historical routes, the probability of having a stronger intellectual DBE effect would be high. The same could happen in relation to the behavioral DBE if the territory under analysis were a destination where water or motor sports were practiced.

## 5. Conclusions

Understanding the influence of brand experience on tourists is crucial to the marketing of a tourist destination [81]. This study corroborates the existing literature considering that DBE is a significant construct with substantial implications regarding visitors' satisfaction and their intentions [7].

The outcomes of this study reveal that visitors are driven by sensory experiences as well as affective experiences to the detriment of behavioural experiences and intellectual experiences.

What attracts visitors the most to this type of territory is nature tourism. However, it is possible to create a more holistic DBE which can trigger various types of experiences in visitors. Firstly, those responsible for territory management must focus on improving the sensory brand experience of consumers in order to increase their satisfaction level regarding the brand. It is important to make a strong impression on consumers' senses. Therefore, it would be interesting to invest in advertising moments or materials that could promote the territory by appealing to visitors' sensations. Secondly, in case those responsible for the territory aim to focus on the brand's behavioural experiences, it would be important to convey the idea that visitors can not only rest, but also perform various types of activities related to hiking, diving, SPA, and birdwatching, among others. To this end, they must support the dissemination and promotion of these activities in the territory. In addition, in case those responsible for the PGNP intend to go further and develop intellectual experiences, it would be necessary to invest in content and disseminate the historical and cultural heritage of the territory, which is still little known.

Therefore, the entities responsible for protected areas in general, and in the PGNP in particular, should place greater emphasis on sensory experiences. Consequently, as we are discussing a protected area, it is necessary to ensure the preservation and conservation

of this space, trying not only to offer the visitors a high-quality experience, but also to preserve the quality of the natural environment on which both the visitors and the host community depend. Underlying this idea, there is a sustainable tourism development which seeks to meet the visitors' needs without compromising the possibilities of the local future generations.

Another issue to be taken into account is the situation of pandemics. The pandemic caused by SARS-Cov-2 immediately promoted not only reflections on its profound effects on tourist activity (e.g., [82]) but also predictions about preferences for travel to destinations such as rural, natural and mountainous areas [83], and for outdoor activities in communion with nature [84]. In the case of Portugal, [85] demonstrated that the pandemic crisis stimulated the increase of domestic tourism and the valorization of rural and natural environments as holiday destinations during the summer of 2020, with emphasis on municipalities located in mountain areas such as Peneda-Gerês, which explains the renewed interest in research on tourists' experiences and loyalty.

Managing the destination brand experience is vital. In fact, both services and context must be approached in a consistent and systematic manner and the specific experiences that ought to characterise the tourism offer have to be wisely selected according to their features [86].

It is therefore important to improve the understanding of the process by which "tourists value their travel experiences so that, based on this knowledge, effective strategies can be defined for the provision of services that can meet [their] satisfaction" [61] (p. 217). With regard to satisfaction, this construct has a very positive influence on intentions to return and intentions to suggest the tourist destination. Interestingly, the mediating effect of satisfaction between DBE and the intention to return and to suggest has support mainly in the sensory and behavioural dimensions. For that reason, given the increase in the number of tourist destinations, the promoters of destinations linked to protected areas should focus on experiential marketing, trying to differentiate themselves from other tourist destinations by the uniqueness, in order to help tourists have a distinct and attractive perceptions of the destination [87].

We consider that this research on destination brand experience, despite corroborating the existing literature, can be further developed, namely in other geographical areas, with other types of target visitors, namely in cities where behavioural and intellectual experiences may be different.

**Author Contributions:** Conceptualization, P.C. and H.M.; data curation, H.M.; formal analysis, H.M.; funding acquisition, H.M.; investigation, H.M., P.C. and N.A.; methodology, H.M. and N.A.; software, H.M.; supervision, P.C. and N.A.; writing—review and editing, H.M., P.C. and N.A. All authors have read and agreed to the published version of the manuscript.

**Funding:** This work was funded by national funds through FCT—Foundation for Science and Technology, I.P., within the scope of reference project no. UIDB/04470/2020.

**Institutional Review Board Statement:** Not applicable.

**Informed Consent Statement:** Not applicable.

**Data Availability Statement:** Not applicable.

**Conflicts of Interest:** The authors declare no conflict of interest.

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
