# Peer review of "Destination Brand Experience: A Study Case in Touristic Context of the Peneda-Gerês National Park"

_sustainability, doi:10.3390/su132111569_

Round 1

Reviewer 1 Report

The manuscript reporting of the case study is interesting, but the authors need to provide a review with some necessary additions:

Line 49: It is important to specify that the integrative models of brand and destination image,  present a consensus practically among researchers about the elements that form the global image of the destination,such as the existence of affective and personal values. What is relevant is that there is no unanimity in the interrelation of the brand and the image of the destination.

Line 51: The brand should be considered as an experience, but this statement should be supported in specific literature that speaks of the need to understand experience for the different stages of the decision-making process of tourism consumers and to find the effect of the different variables in making decisions about a certain tourist destination. PRAYAQ AND RYAN (2011), "Antecedents of Tourists' Loyalty to Mauritius: The Role and Influence of Destination Image, Place Attachment, Personal Involvement, and Satisfaction", Journal of Travel Research, Vol. 50, no. 3.

Line 60: It is recommended to incorporate basic information on the protected area under analysis and, in general, on the characteristics of the northwest and the Luso-Spanish border area. This will serve as a support of knowledge to thereader. The incorporation of a basic mapping of the situation of the study area is requested.

Line 78: The experienced branding applied to tourism has been in clear development for at least a decade. It is suggested to consult manual: World Tourism Organization and European Tourism Commission (2011), Manual on branding of tourist destinations, UNWTO, Madrid, DOI:  https://doi.org/10.18111/9789284413706

Line 98-99: There is talk of an extensive literature review, but it is not specified whether the previous scale examples have been considered for the formulation of the hypothesis or if the approaches are new. Clarification is requested on this, including specific information on the sources of information consulted and used as a reference for the configuration of the four-dimensional scale.

Line 107: Apart from what has been briefly commented, no reflection is provided on the contribution of the proposed four-dimensional scale. This reflection, although it should not be extensive, should clearly explain the benefits that research understands and hopes to generate with its use. Likewise, a reflection is required on the degree of adequacy of this proposed scale to the study area, and whether this methodology is only usable in the case of the analyzed National Park or can be exported to other environments.

Line 140: In addition to the advantages cited over the brand and brand consumption focused on tourism, the authors have ignored the spatial and social dimension in the development of the introduction. It is recommended to reformulate this part on Brand Experience applied to tourism so that the vision of the role of brandig strategy as a favoring of the economic, social and cultural development of space isintroduced.

Line 153: The authors recurrently express the need to incorporate sensory and affective experiences among the factors to be studied in the behavior of tourism branding, without making any development on these aspects. Due to the indeterminacy and subjectivity that these phenomena entail, a greater concreteness in their definition, explanation and development is requested.

Line 216-221: When reading this paragraph, which is interpreted as a theoretical conclusion of the manuscript, a background of concept is revealed that makes it difficult for the reader to differentiate between branding and marketing actions in tourism.

Methodology: The questionnaire methodology is clear. Even so, it is necessary to include a higher level of detail about the design of surveys and the types of questions used, which are not correctly categorized in the added tables because only the results obtained from the average of answers are provided.

The format of the questions is also unclear: Not all survey questions are created equal. For the design of surveys, both closed questions and open questions can be used, which can provide limited answers or expand the information with answers obtained from the respondents' own words.

Conclusions: It would be necessary to identify, for example, what aspects have been worked on for the development of the tourism brand in study, and by what direction the tourism-space planning policy should go to achieve the objectives set out in the premises section of the text.

Although the summary refers to the impact that pandemics have, for example, on the way people travel, this reflection section does not reflect the authors' idea of how these phenomena can affect when changing or modifying the branding strategies implemented.

Author Response

Thank you for Your comments and the opportunity to resubmit our manuscript. Please see the attachment.

Reviewer 2 Report

The article is an interesting study with good scientific sundeness. The structure is logical but incomplete. There is no scientific discussion that would summarize the obtained results against the background of other studies, when citing literature sources. Thanks to this, it would also be an indication for other researchers in comparative studies. It is necessary to clarify the research hypotheses, meaning that the evidence relates to this particular case study. The methodology does not raise any objections. It is also desirable to develop literary studies at the beginning of this article. Overall, it is an interesting study, worth publishing after taking into account the suggested changes and additions. 

I recommend:

http://dx.doi.org/10.18276/ept.2015.3.31-07

Author Response

Thank you for Your comments and the opportunity to resubmit our manuscript. Please see the attachment.

Regards!

Reviewer 3 Report

Although, the study makes a nice contribution to the subject it seems incomplete. Results are interesting.
-The paper lack of an “added value”, also indicated by a short “Discussion".
I would encourage the authors to consider improving their analysis on a more comprehensive critical discussion on the issue. Really, In the section 4 there is not a discussion but only results.
-The 'Discussion' lacks greater rigour and the concluding sections of the paper, particularly those dealing with contributions to practice, limitations, and directions for future research, seem all too thin. The treatment of implications for practice are particularly thin though the author(s) do try to examine the implications of their work for the existing literature.

Author Response

(The authors gave the same response as above.)

Reviewer 4 Report

It is a work on topics currently very much in vogue, but whose conclusions are mainly quantitative, not bringing scientific news on the topics addressed or improvements to the destination object of the case study.

The biggest problem with this article is the structure of it.

A literature review is presented without a well-defined conceptual scheme.

The detailed explanation of methodology is a fact. However, it is based on a survey carried out as early as 2016, which is in addition to the fact that it was applied by persons not directly related to the study.

The object of study, the brand Peneda-Gerês National Park, and the PGPN himself, is not properly presented or contextualized.

The results are presented in an unclear way because they are not easily understandable, so they do not benefit the tourism sector or the users of the Peneda-Gerês National Park. The paper almost presented quantitative results.

The case study is therefore very small. In about two and a half pages, to which adds a conclusion that nothing brings again.

It refers a lot to the literature on the themes treated, but no comparative studies are presented.

There is the assumption by the authors that the data provided by the surveys and the methodology used and applied are sufficient.

Author Response

(The authors gave the same response as above.)

Round 2

Reviewer 1 Report

The authors have followed the recommendations made in the first round of review, which have significantly improved the different sections of the text. In its current form, the manuscript has well-defined objectives and hypotheses, responds clearly to them through the methodology followed and suggests some relevant conclusions and opens the debate on the subject, very much in vogue today in tourism.

Reviewer 2 Report

I accept the changes and additions introduced by the Authors. Recommend the article for publication.

Reviewer 3 Report

At this version, the paper is improved. 

Reviewer 4 Report

Accept in present form